

# Spawning of French grunts, *Haemulon flavolineatum*, in recirculating aquarium systems

Leah Maurer[1], Matthew Dawson[1], Larry Boles[1], Stacy Knight[1] and Andrew Stamper[2]

[1] Epcot's The Seas with Nemo & Friends®, Walt Disney World Resort®, New College of Florida, Lake Buena Vista, FL, USA

[2] Epcot's The Seas with Nemo & Friends®, Walt Disney World Resort®, Lake Buena Vista, FL, USA

## ABSTRACT

**Background**. Because the French grunt, *Haemulon flavolineatum*, is an ecologically important reef fish prized by both recreational anglers and public aquariums, the wild population requires limits on harvests. Yet, the environmental conditions conducive for French grunt spawning in aquarium settings is not well understood. Therefore, the goal of this study was to document the conditions leading to voluntary spawning and the number of eggs produced by French grunts without the use of hormones or artificial insemination.

**Methods**. We hypothesized and verified that it is possible for French grunts to spontaneously spawn in human care. Forty individuals were collected around the Florida Keys and haphazardly stocked in five recirculating seawater systems each containing two 250-L circular tanks. Over the course of 87 days, eggs were collected daily from each system and environmental parameters were monitored.

**Results**. Total daily number of eggs released ranged from 0 to 207,644 eggs. Of the observed environmental parameters, temperature and alkalinity had the greatest impact on number of eggs released. This study demonstrates that it is possible for French grunts to reproduce in captivity with little environmental manipulation, thus an ideal candidate to culture for the zoo/aquarium industry.

Corresponding author
Leah Maurer,
Leah.M.Maurer.-ND@disney.com

## INTRODUCTION

While a majority of aquaculture production worldwide is devoted to food production, ornamental fish production is the fourth largest sector in the United States aquaculture industry (*Tlusty, 2002*). The United States of America is the single largest importer of ornamental fish; the aquatic ornamental industry is estimated at about 60 million US dollars in 2012 (*Bassleer, 2015*). Unfortunately, more than 90% of marine fish in the ornamental industry are collected from the wild (*Chapman et al., 1997*; *Monticini, 2010*) stressing the need for captive breeding of marine ornamental species in order to supplement or replace the supply of wild caught specimens, and potentially improve wild stocks.
In recent years, grunts of the genera Haemulon have attained a high economic importance (*Weiler & Suarez-Caabro, 1980*). The species is valued by both recreational anglers and public aquariums around the world thus requiring limits on wild harvests. Anglers prize grunts as a recreational fishery while the grunts' schooling behavior, color and hardiness make them an ideal candidate to display in public aquariums. A better understanding of French grunts spawning in recirculating systems can lead to more aquaculture thus alleviating the pressures on wild populations.

Grunts of the genera Haemulon are distributed in tropical and sub-tropical climates along the western Atlantic and the eastern Pacific oceans (*Gaut & Munro, 1983*). French grunts, *Haemulon flavolineatum*, are ecologically important because the species creates trophic links between reef and seagrass environments (*Nagelkerken et al., 2008*). Post-larval settlement occurs in mangroves and seagrass beds approximately fifteen days after fertilization (*McFarland et al., 1985*). Juveniles remain in these coastal habitats feeding primarily on macro-invertebrates until they reach sub-adult size and migrate offshore (*Verweij et al., 2006*; *Verweij & Nagelkerken, 2007*). French grunt adults generally inhabit coral reefs and rocky outcrops, often in schools, but migrate to surrounding grass beds and sand flats at dusk to forage solitarily (*Burke, 1995*).

While much of their ontogenetic biology has been studied, little is known about the reproductive life history of the French grunt specifically the conditions leading to spawning, the number of offspring and frequency of reproduction. Investigations describing spawning activity in the French grunt have been based on gonad tissue examination (*Gaut & Munro, 1983*) and settlement rates (*McFarland et al., 1985*; *Shulman & Ogden, 1987*). Overall, spawning in the French grunt is not well documented or described thus more information is required to successfully reproduce this species in human care. Spawning in other Haemulidae species, however, has been successfully documented such as in the common or white grunt, *H. plumieri* off the coasts of Florida (*Moe, 1966*), Puerto Rico (*Erdman, 1977*) and Venezuela (*Palazón-Fernández, 2007*). In artificial systems, successful spawning was documented in the sweetlips grunts, *Plectorhinchus vittatus* (*Leu et al., 2012*) and *P. pictus* (*Horike & Kawahara, 1982*).

In August 2012, French grunts were collected from the wild to display at Walt Disney World®'s The Seas with Nemo & Friends®. French grunts are favored among aquariums due their schooling behaviors and bands of fluorescent blue set amidst a vibrant yellow body. Before introduction into the aquarium, the French grunts began spawning of their own volition without the use of hormones or artificial insemination. This provided an opportunity to study spawning in human care which is not well understood. We hypothesized that it is possible for French grunts to spontaneously spawn in human care. Egg production was observed for 87 days. The objective of this study was to document the environmental conditions (temperature, dissolved oxygen, pH, salinity and alkalinity) as well as the sex ratios leading to voluntary spawning and the number of eggs produced by French grunts in recirculating aquariums.

**Table 1  Male to Female (MF) sex ratios of *Haemulon flavolineatum* housed in five recirculating systems from January to March.** Fish were grouped to minimize aggression and sex was determined by gamete sampling primarily via a catheter.

| System | 1 | 2 | 3 | 4 | 5 |
|---|---|---|---|---|---|
| M:F ratio for Tank A | 2:2 | 1:3 | 1:3 | 0:4 | 0:4 |
| M:F ratio for Tank B | 2:2 | 2:2 | 3:1 | 1:3 | 2:2 |
| Total M:F ratio | 4:4 | 3:5 | 4:4 | 1:7 | 2:6 |

## MATERIALS & METHODS

### Brood stock maintenance

Wild French grunts were collected around the Florida Keys, USA by a commercial supplier (Dynasty Marine, Marathon, FL, USA) and transported to The Seas with Nemo & Friends® in 2012 to supplement the population of fish on display. Upon arriving, the fish underwent a 30-day quarantine period. All procedures described herein were reviewed and approved by Walt Disney World's Animal Care and Welfare Committee (IR #1301). Quarantine treatments included therapeutic copper sulfate bath, 0.18–0.2 mg/L (Sigma-Aldrich Corporation, St. Louis, MO, USA) for 21 days. Additionally, two 6-hour baths of praziquantel, 2mg/L (Sigma-Aldrich Corporation, St. Louis, MO, USA), were given 10 days apart and fendbendazole, 25 mg/kg (Panacur®, Merck & Company, Kenilworth, NJ, USA) was administered via diet for three consecutive days and repeated seven days later.

After the quarantine period, 40 fish were moved into five 545-L recirculating systems until the fish were ready to be placed on display. Each 545-L system was comprised of two 250-L circular tanks designated as Tank A and Tank B, and each pair shared recirculating water (Table 1). French grunts are known to display intra-species aggression and this behavior can be exacerbated in small environments. To minimize potential aggression and improve animal welfare, four fish of similar size were housed together in each tank. Prior to the experiment, behavior was monitored and fish were moved around between tanks until aggression was minimal or subsided. Ultimately, System One housed two males and two females in both Tank A and Tank B delivering a four Male to four Female ratio for the whole paired system. System Two housed one male and three females in Tank A while Tank B housed two males and two females delivering a three Male to five Female ratio for the whole paired system. System Three housed one male and three females in Tank A while Tank B housed three males and one female delivering a four Male to four Female ratio for the whole paired system. System Four housed no males and four females in Tank A while Tank B housed one male and three females delivering a one Male to seven Female ratio for the whole paired system. System Five also housed no males and four females in Tank A while Tank B housed two males and two females delivering a two Male to six Female ratio for the whole paired system.

In addition to the two 250-L circular tanks, each paired system contained a canister filter, and a sump. Saltwater was prepared using artificial salt mix (Instant Ocean®, Spectrum Brands Inc, Atlanta, GA, USA) and biological filtration was established using liquid bacteria starter (Frit-Zyme®, Fritz Industries Inc, Mesquite, TX, 75149). Water

flowed from the tank into the sump containing a 100 μm prefilter (Coralife®, Central Garden & Pet Company, Franklin, WI, USA), the water was filtered through a wet-dry trickle filter then a 20 μm pleated canister filter (Lifeguard Aquatics®, Cerritos, CA, USA) before returning to the tanks. A 25 W UV sterilizer (Aqua Ultraviolet®, Temecula, CA, USA) was supplemented to each system for at least 48 h per week. No carbon filtration or foam fractionation was used to treat the system water.

Fish in all the tanks were fed omnivore aquatic gel (Mazuri®, Richard, IN, USA) approximately three percent of the total body mass of the population of the tank. Feedings were divided into two portions given in the morning and in the afternoon. One hour after every feeding, leftover food was removed via net.

### Environmental conditions

Temperature ($\sim$21$-$29 °C), dissolved oxygen (DO) (6.1–7.4 mg/L), and pH (7.7–8.5) were recorded daily for each system where as salinity (29-36 ppt) and alkalinity (2.2–7.0 meqa/L) were recorded sporadically due to limited access to required meters. Sodium carbonate and sodium bicarbonate were added as needed to maintain pH and alkalinity. Each system received a 30% water change every 14 days. Artificial light ($\approx$600 lux) was supplied by 20, 32-W fluorescent bulbs at a 12 L: 12 D photoperiod.

This study was an experiment of opportunity. Before the fish were moved on display, spawning was observed. Once repeated spawning was confirmed, daily environmental monitoring and quantification of egg production ensued for a total of 87 days from January 1, 2013 to March 28, 2013.

### Sex determination

The sex of each fish was determined by gamete sampling primarily via a catheter. Fish were anesthetized prior to catheter insertion at 100 ppm Tricaine Methanesulfonate (Tricaine-S®, Western Chemical Inc, Ferndale, WA, USA). A two-inch segment of 3.5RR Fr catheter tubing (Kendall™, Tyco Healthcare Group LP, Mansfield, MA, USA) slid over a 22-gauge needle attached to a one-cc syringe was gently introduced into the genital pore of the fish. Sterile saline was used to flush the genital duct and a slight suction was applied to draw eggs or milt into the catheter. In cases where sex determination via catheter insertion was difficult, mainly occurring in males, strip spawning was effective in collecting gametes. A total of 18 males and 22 females were determined.

### Egg Collection

Because of the sudden opportunity to study spawning in these Grunts and the nature of the paired systems, an unusual method of collecting eggs was devised that collectively pooled eggs from both tanks A and B. Eggs were collected by a 100 μm mesh prefilter during the night and were gently removed from the prefilter using a metal spatula the following morning. Daily egg volume was quantified by concentrating collected eggs in a graduated cylinder with water decanted. Before the start of the monitoring period, mean egg diameter of $0.95 \pm 0.006$ mm (mean $\pm$ SE) was calculated by measuring 30 haphazardly selected eggs across two of the tanks using microscope camera software (Infinity Capture®, Lumenera Corporation, Ottawa, ON, Canada); microscope lens provided scale to analyze

egg diameter. Any daily or individual fluctuations in egg size was thought to be minimally impactful to mean egg diameter due to (*Chambers & Leggett, 1996*) which concluded that egg size within a population varied about 4%. The average was used to calculate mean egg volume ($\frac{4}{3}\pi * r3$), $r = 0.47$. The number of eggs per mL (N) was calculated: N = $\frac{1}{0.00044}$, mean egg volume (mL) = 0.00044. The total number of eggs released for each system was then determined by multiplying N by the total egg volume collected [rounded to nearest 0.5 mL].

Fertilization or egg viability was not monitored in this study. The purpose of this study was to document the environmental conditions leading to voluntary egg production. However, all eggs collected were given to the University of Florida's Tropical Aquaculture Laboratory for further research (*Wittenrich et al., 2017*).

### Statistical analysis

The goal of this study was to document the conditions leading to voluntary spawning and ascertain the degree to which environmental parameters affected the number of eggs produced. Our ultimate goal is to maximize production for future French grunt culture in the zoo/aquarium industry. We used R statistical software (R core team 2020) to perform statistical analyses. First, a linear model was performed to determine differences between each of the five systems followed by post-hoc Tukey test. To determine which environmental parameters affected egg production, we created a list of *a priori* models, analyzed with general linear-mixed models using the R package lme4 (*Bates et al., 2017* p. 4). We then used AICc values to rank and compare the a priori models to determine the model that best fit the data using. We used the AICcmodavg package in R to perform the model selection (*Mazerolle, 2016*). Finally, to determine statistical significance of the top model we used the R package lmerTest (*Kuznetsova et al., 2016*). The dependent variable was the number of the eggs and the environmental parameters were independent variables.

## RESULTS

Spawning occurred on average $51.7 \pm 3.5$ days out of 87 days for all five systems (51.7% of the time) from January to March (Fig. 1). Egg production occurred only overnight (1700 to 0800 h) as no additional eggs were discovered on the prefilters after the morning egg collection. Even though spawning was only monitored from January to March, the bulk of egg production occurred in March.

Mean number of eggs released for the systems ranged from 5562 to 16699 eggs over the 87-day period with the highest mean number of eggs released occurring in system two and the lowest number of eggs released occurring in system three (Fig. 2). The overall mean number of eggs released was 9,232 eggs for all five systems (Fig. 3). According to linear model and Tukey post hoc analysis, number of eggs released was significantly different between the systems ($F = 2.39$, $df = 4$, $p = 1.2E - 06$, $\eta^2 = 0.08$). The general linear mixed model that best fit our data included only temperature and alkalinity as fixed effects, and included date and system as random effects. Both temperature and alkalinity had a significant effect on egg production ($t_{10.24} = 2.597$, $P = 0.03$ and $t_{21.54} = 2.592$, $P = 0.02$; t values determined using t-tests use Satterthwaite approximations). The top

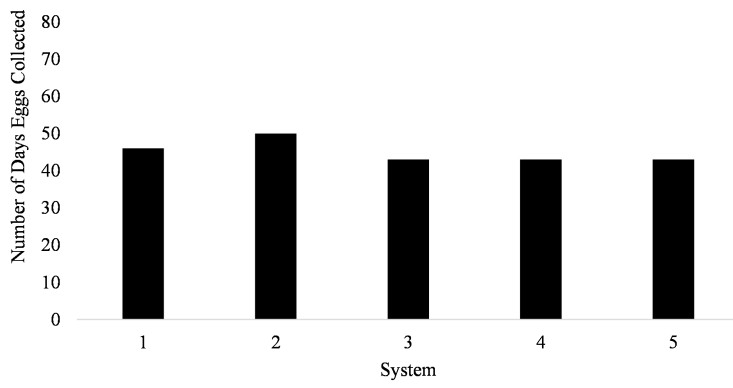

**Figure 1** Number of days *Haemulon flavolineatum* eggs were collected from five paired recirculating systems over a total of 87 days of observation.

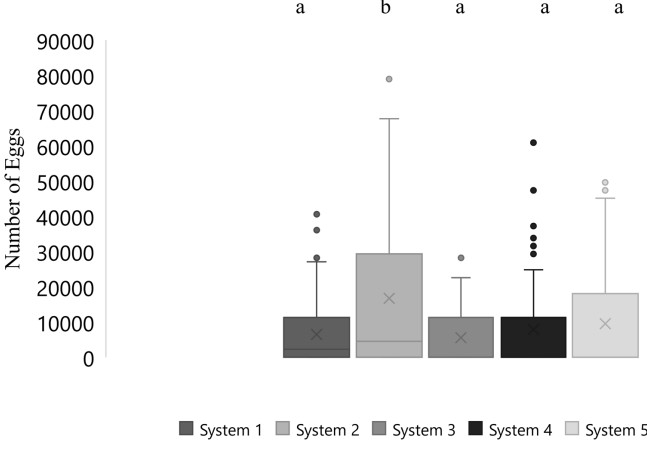

**Figure 2** Boxplot of daily number of eggs collected of *Haemulon flavolineatum* housed in five recirculating systems from January to March. Letters indicate significance according to linear model and post hoc Tukey test.

model accounted for 51% of the Akaike weights with a delta AICc of 2.67 compared to the next best model. Furthermore, the next best models included more environmental parameters, although only temperature and alkalinity were ever significant.

A positive trend was observed in mean daily number of eggs released and mean daily temperatures (Fig. 4A, $R^2 = 0.18$) although the relationship was not significant. Additionally, a positive trend was also observed in mean daily number of eggs and mean daily alkalinity values (Fig. 4B, $R^2 = 0.14$) but the relationship was not significant.. The bulk of egg production occurred in March (monthly mean temperature 27.5 °C and alkalinity

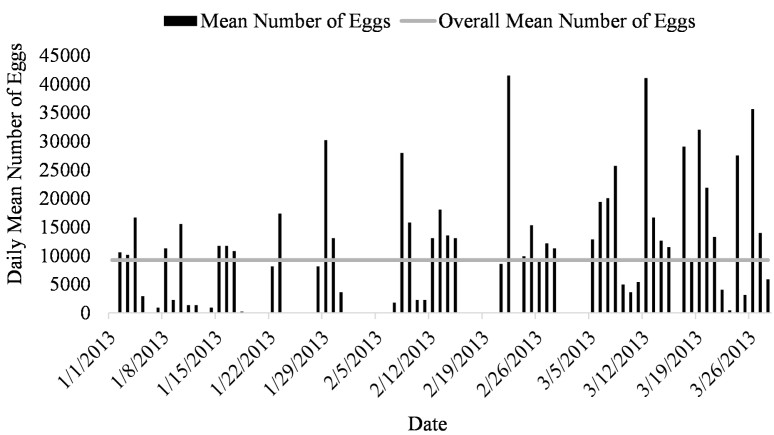

**Figure 3** Daily mean number of eggs (black) and overall mean number of eggs collected (gray line) of *Haemulon flavolineatum* housed in five recirculating systems from January to March.

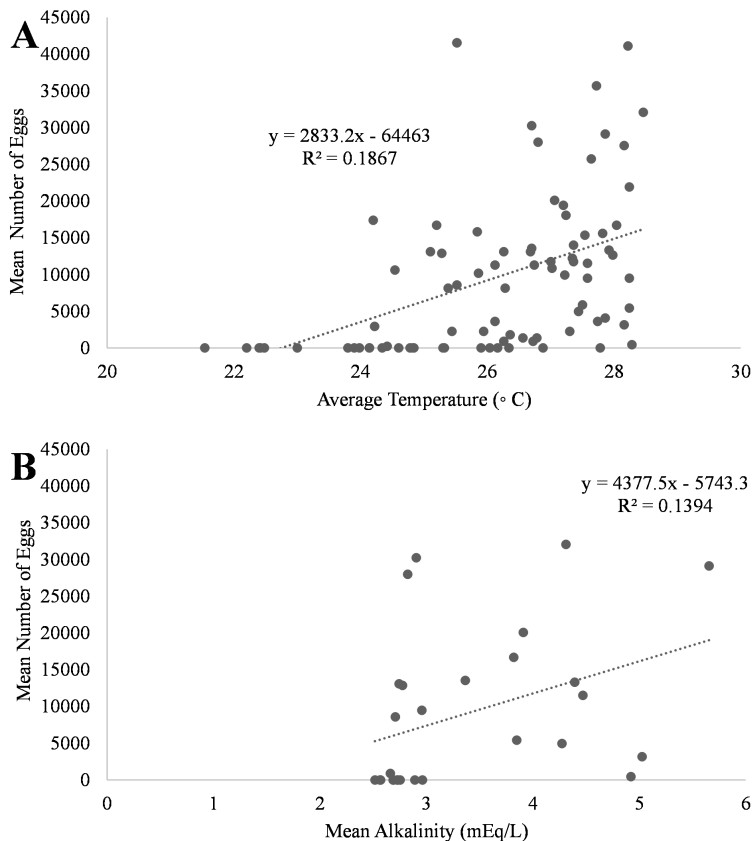

**Figure 4** Relationship of mean number of eggs of *Haemulon flavolineatum* and (A) mean temperature and (B) mean alkalinity in five recirculating systems from January to March.

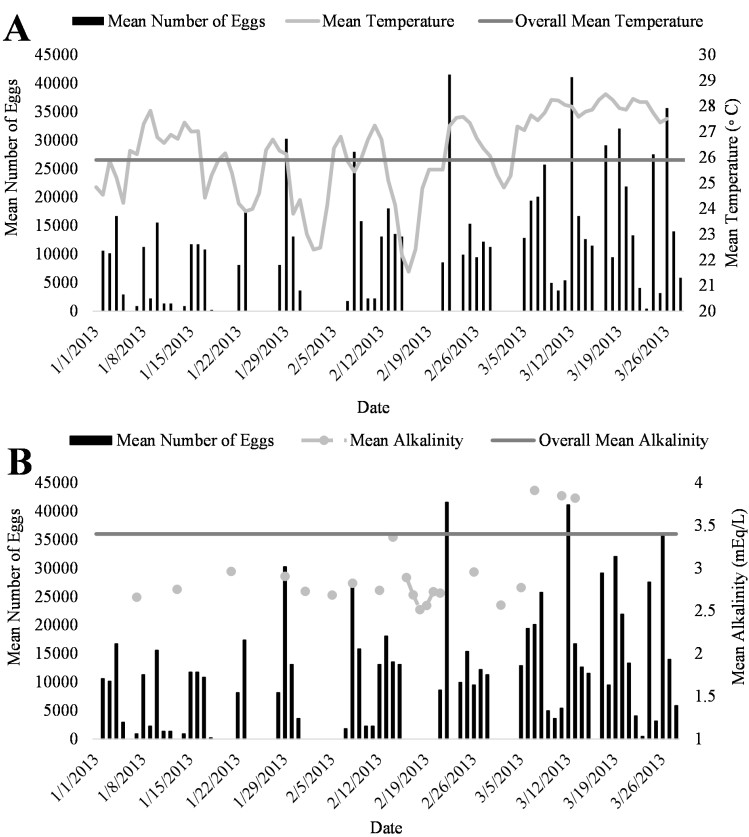

**Figure 5** Relationship of mean daily number of eggs (black) of Haemulon flavolineatum and (A) mean temperature (light gray) with overall mean temperature (dark gray) and (B) mean alkalinity (light gray) with overall mean alkalinity (dark gray) in five recirculating systems from January to March.

4.2 mEq/L) which had the highest mean temperatures and alkalinity values during the study (Fig. 5).

Egg production was observed in each system regardless of female to male ratio. Due to the nature of the paired systems, the number of eggs had to be collectively pooled from both tanks A and B. The number of eggs in relation to M:F ratio had to be analyzed per system and not per individual tank. However, a negative trend was observed in mean daily number of eggs as the male to female ratio reached one (Fig. 6). The highest number of eggs was observed when the population was skewed more to females.

## DISCUSSION

Previous studies investigating the reproductive life history of French grunts were strictly based on examination of gonad tissue in wild fish (*Gaut & Munro, 1983*) and analysis of settlement rates (*McFarland et al., 1985*; *Shulman & Ogden, 1987*) but they did not illustrate the conditions leading to egg production. To successfully reproduce French grunts in human care, the environmental conditions leading to spawning without the use of hormone injections or artificial insemination must be better understood. Voluntary

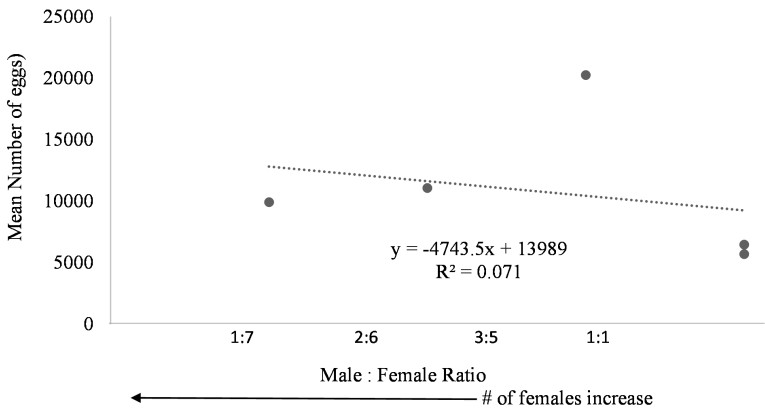

**Figure 6** Mean number of eggs in relation to male to female ratio of *Haemulon flavolineatum* housed in five recirculating systems from January to March.

spawning in haemulids in managed care has been limited mainly to the genera *Plectorhinchus* (*Leu et al., 2012*; *Horike & Kawahara, 1982*).

The present work outlines the environmental conditions and sex ratios that led to spawning in French grunts held in recirculating aquarium systems. However, some limitations should be noted. A larger sample size would have been preferred to improve statistically reliability and power. Unfortunately, the systems could not be modified to prevent pooling of eggs so any data that could have been collected from individual tanks and associated sex ratios was lost. The number of eggs had to be analyzed on a system level. Additionally, because the fish were originally intended for display purposes, the fish were housed in groups that showed minimal aggression to sustain healthy vibrant physical appearances regardless of sex. This caused the sex ratios for each system to be either equal or skewed towards females. A dominance of females is common among Haemulon species (*Gaut & Munro, 1983*; *Palazòn-Fernandez, 2007*). This lack of males in a tank may have interfered with the reported results because it is unknown how the presence of males and their pheromones affect egg production. Furthermore, the experiment only last 87 days. Because the fish were intended to be put on display, we had a limited amount of time to collect data thus limiting improvements or modifications to methodology.

Environmental cues such as photoperiod and temperature are known to initiate spawning for many tropical fishes in both wild and tank-raised settings (*Holt & Riley, 2001*; *Robertson, Green & Victor, 1988*). With a constant artificial photoperiod, temperature showed to have a large effect on egg production in this study. Even with the small sample size and time period, the data presented here indicates that French grunts can spawn continuously when water temperature is between 21−29 °C with greater number of eggs released in the months of March when water temperatures were at their highest. Water temperatures (21−29 °C) observed during this study were similar to those observed by *Saksena & Richards (1975)* (24.2–27.5 °C) while collecting wild eggs of other Haemulon species with plankton nets.

The present work did not determine the true spawning season of French grunts because photoperiod was held constant and natural seasonal changes, with the exception of ambient temperature, were not replicated. However, this strategy may not be true of all populations found in the Western Atlantic. Better understanding of the peak spawning season could help maximize egg production of French grunts. Additionally, the continuous spawning observed in the present study, even in cooler water temperatures (21−24 °C), supports observations from *Gaut & Munro (1983)* and *McFarland et al. (1985)* that wild French grunts, particularly in tropical habitats, spawn year-round.

One mechanism that would explain year-round egg production in the French grunt is asynchronous oocyte development within the female. If oocytes develop in different cycles concurrently within the same ovary, a French grunt female could produce eggs throughout the year at a low abundance as not to expend an excessive amount of energy. This reproductive strategy is typically found in serial spawning fish and has been reported in multiple *Haemulon* species including the common or white grunt (*Palazón-Fernández, 2007*). Future studies examining the ovarian tissue of mature French grunt fish to determine variable oocyte development would help to confirm this hypothesis.

In addition to their natural spawning season, the reproductive behaviors of French grunts are not well documented. No mating behaviors were analyzed in this study. Tank mate chasing and aggression were observed during the day, but future studies using red light and video analysis must be conducted to capture spawning behavior at night. It is documented that French grunts are pelagic spawners with males and females broadcasting sperm and eggs into the water column simultaneously thus allowing fertilization to occur (*Gaut & Munro, 1983*).

For this experiment, egg production only occurred overnight (1700 to 0800 h). Time of day can be a very important factor for spawning (*Sancho, Solow & Lobel, 2000*). Although it is well known that some tropical marine fishes spawn during the day, the majority of species spawn at night or at dusk (*Delsman, 1930*; *Bapat, 1955*; *Allen, 1972*; *Mariscal, 1972*; *MacDonald, 1973*; *Moyer & Bell, 1976*; *Hobson & Chess, 1978*; *Ross, 1978*; *Lobel, 1978*; *Thresher, 1982*; *Colin & Clavijo, 1988*; *Colin & Bell, 1991*; *Robertson, 1991*; *Sancho, Solow & Lobel, 2000*). Spawning during the night reduces the threat of predation on fish (*Hobson, 1975*; *Hobson & Chess, 1978*) and may be an adaptation to reduce egg predation (*Johannes, 1978*).

In this study the relationship between maternal condition and egg production was unclear due to the small sample size and paired systems. Even though weight is usually the better predictor of egg production, this study showed that the highest number of eggs was observed in system two which had one of the lowest mean female weights (91.6 g) and the largest mean female fork length (16.0 cm). *Koops, Hutchings & McIntyre (2004)* demonstrated that length-based regressions can over-estimate correlations between maternal body condition and fecundity, suggesting that the effect of body condition on egg production may not be as universal or as biologically important as previously thought.

Even though system four and five contained tanks with only female fish, egg production did not appear to be hindered suggesting that there may be a chemical cue stimulating spawning because the females couldn't see the males present in the system. This

phenomenon is not uncommon among fish. Male chemical cues resulted in higher spawning rates for angelfish (*Chien, 1973*) and gourami (*Cheal & Davis, 1974*) when the male fish were not present. Further studies are required to determine the influence of male chemical cues upon the spawning rate of female grunts. Additionally, the absence of males could have had a dramatic effect on the spawning behavior of the females. Further research with a larger sample size examining the ideal sex ratio for egg production is required; however, in this study more eggs were produced when population was skewed toward females with highest egg production at male to female ratio of 3:5. Understandably, the more females present, the more eggs are to be expected.

Because the French grunts in this study were wild-caught, it was impossible to determine the age of female grunts when voluntary spawning was first observed. However, eggs collected during this study were transferred to University of Florida's Tropical Aquaculture Laboratory for larviculture research. When F1 generation grunts were held in the same systems as this study they started spawning at 15 months old. Studies on French grunt larval development, growth, and survivorship were conducted to establish a larval rearing protocol (*Wittenrich et al., 2017*).

## CONCLUSIONS

In conclusion, this study verified that French grunts can spontaneously spawn in human care and documented conditions leading to voluntary spawning. It is expected that these results will help to promote successful, sustainable commercial production of French grunts decreasing the pressures on wild populations. Because this study was an experiment of opportunity, we were limited in time and methodology. Our results show that egg production in French grunts is strongly dictated by water temperature and alkalinity, influenced, albeit loosely, by the number of females (when similar in size). It is our recommendation that French grunts should be reared in water temperatures between 26−28 °C with alkalinity ranging from 3.5–4.2 mEq/L and the populations should consist of about 60% females to encourage more egg production. Results from this work demonstrate that French grunt spawning is able to occur in captivity with little environmental manipulation and thus an ideal candidate for aquaculture.

## ACKNOWLEDGEMENTS

The authors would like to thank Nicole Uibel, Melyssa Allen, Courtney Duff, and Alex Novarro for their assistance with data collection and fish husbandry. Special thanks to Dr. Matthew Wittenrich for his technical guidance and support, to Charlene Burns for all her veterinary aid and expertise, and Dr. Zak Gezon for his statistical assistance. We also would like to thank to Amber Thomas for her comments and constructive criticism during manuscript revisions. This study was conducted opportunistically in conjunction with the Rising Tide Conservation project at University of Florida's Tropical Aquaculture Laboratory.

### Funding

The authors received no funding for this work.

### Competing Interests

All authors either worked at Walt Disney World Resorts during the time of research or are still working at Walt Disney World.

### Author Contributions

- Leah Maurer analyzed the data, prepared figures and/or tables, authored or reviewed drafts of the paper, and approved the final draft.
- Matthew Dawson performed the experiments, analyzed the data, authored or reviewed drafts of the paper, and approved the final draft.
- Larry Boles and Stacy Knight conceived and designed the experiments, performed the experiments, authored or reviewed drafts of the paper, and approved the final draft.
- Andrew Stamper conceived and designed the experiments, authored or reviewed drafts of the paper, and approved the final draft.

### Animal Ethics

The following information was supplied relating to ethical approvals (i.e., approving body and any reference numbers):

Walt Disney World's Animal Care and Welfare Committee reviewed and approved all procedures (IR #1301).

### Data Availability

The raw data are available in the Supplementary File.

### Supplemental Information

Supplemental information for this article can be found online at http://dx.doi.org/10.7717/peerj.9417#supplemental-information.

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
