# Peer review of "Spawning of French grunts, Haemulon flavolineatum, in recirculating aquarium systems"

_PeerJ, doi:10.7717/peerj.9417_

## Round 0.1 · original submission · Major Revisions

Thank you for your submission to PeerJ. Two expert reviewers and I have now read and reviewed the manuscript and provided comments for improvement. The reviewers believe that there some important information regarding reproduction in the French grunt, and given appropriate framing and context, the manuscript should be published. However, it will need to undergo major revisions before it can be published. Please address each reviewer comments upon resubmission, as well as my own comments. I have commented primarily on statistical validity, which is currently not adequately explained, and likely needs to be completely redone using appropriate statistical methods. Reviewer 2 also made some points about the tank design and how these different sex ratios and tanks need to be incorporated in the analysis.

Additionally, both reviewers and I have commented that the manuscript needs to be reframed, with a clear explanation of the hypotheses/objectives and importance of the work and how it fills gaps in the literature. Further, the utility of the results needs to be clarified, which might involve recommendations for the future.

Specific manuscript comments:

Line 18 - Replace “since” with “because” in this context (here and throughout the manuscript).

Line 50 - ¬Need a citation here – it reads a bit strange. Perhaps it should be changed to “French grunts are ecologically important because….”
How is this species of ecological importance? The next few sentences state some of the ecology of the species, but not its ecological importance. Please revise.
The study species is now introduced, but it is unclear why this is the species of the study.

Line 60-61 - Give more detail as to what is actually missing in terms of knowledge. And then explain how this research fills some of those gaps. The manuscript needs a clear hypothesis/aims/research questions statement in the introduction to inform the reader of the purpose of this manuscript.

Line 70 – Identify the conditions.

M&M – Subheadings would help readability

Lines 83-87 – Please clarify again for the readers – there are 5 pairs of tanks, and each pair share recirculating water. It isn’t super clear throughout you’re your raw datasheets. Also please explain the sex ratios per tank in paragraph form with how many replicated there were per ratio. The table is helpful, but it also needs to be clear in the methods section.

Line 119 – A subheading or opening sentence here to explain that this method was used to estimate the number of eggs laid would be useful.

Lines 122 – Were the 30 eggs collected daily, or was it once over the course of the monitoring period? Across multiple tanks? How might daily or individual fluctuations impact these results?

Line 128 – Clarify the statistical software used. Be clearer in the dependent variable vs independent variables, and if the model used was additive or multiplicative. What is a system? - line 130. What environmental conditions were used? Does fecundity take into account the number of females present in the tank? Because presumably if all females are releasing eggs there should be more eggs in the tank with more females.
Mean fecundity was the dependent variable? Does that mean you only included 10 data points in the analysis because fecundity was averaged over time? That is not enough to assess the statistical importance of so many independent variables.
There are a lot of holes in the statistical analysis section – the authors really need to be clear about what they did, what the data was, and what the exact independent variables were. A linear mixed effects model is likely more appropriate, and date and tank (as a random effect) should be included in the model. It is not statistically sound to perform different ANOVAs for each independent variable of interest, but rather perform model comparisons with multiple different variables. However, I am unsure there is enough data to do this adequately, and there certainly isn’t enough data if the average fecundity per tank is estimated. Additionally, day does need to be included in your analysis because, presumably, these females are not regenerating eggs within the approximately 90-day time frame, and it would take more time to develop egg reserves. Therefore, if they are releasing their eggs there might be a finite number within each female – and therefore date is an important variable to include.
Because I am unclear on the statistical analysis performed -and if my assumption that individual ANOVAs were performed for each independent variable, then the statistical results are not reliable - I will not comment on the statistical results, and await the revised version of the manuscript.

In reading the manuscript, I am having issues with the terms used to describe egg release. Fecundity is perhaps the wrong word choice because it typically applies to an individual and often references number of fertilized eggs or at least egg viability. There was no test for fertilization success or egg viability. The authors are only measuring the number of eggs released and not true fecundity
Furthermore, eggs are seeming released regardless of male presence (line 144), and fertilization success was not recorded nor egg viability assessment conducted. Volitional spawning is typically a term used with artificial reproduction and hormone injections, where the eggs are collected to be artificially fertilized. Volitional spawning without the artificial semination does not yield offspring and could be an indication that females are in stress, or avoiding being egg bound. (My expertise is not in fish spawning but frogs, so forgive me if these assumptions are incorrect). Because you didn’t measure fertilization, or egg viability and females released eggs in the absence of males where no fertilization could take place, what is the importance or utility of this study? The purpose for the females to release the eggs might not be for reproductive reasons, but rather because of stress, or to avoid being egg bound. Make sure in the revision to be very clear about the utility of these results.
Note: However, if there was at least 1 male in the system of 2 tanks (which seems to be the case according to table 1), and if lines 214-216 are true, then the females could be spawning based on chemical cues in the water. And this would need to be more fully explained in the manuscript.

Lines 180-181. This was mentioned in the intro but need to be clear about the gaps and why these are not appropriate ways or not exhaustive ways of investigating the reproductive life history.

Lines 212-214. The use of red lights and video to conduct a behaviour study could be attempted, but it wasn’t in this case. This reads as an excuse now – place it instead as future research. Fertilization and egg viability assessment would also be a good next step.

The discussion needs to be modified to explain the utility of this study. There should be some discussion of limitations, and proposals for future research, but it needs to comment on spawning recommendations, and describe how this study can be used by others. Further, it will likely need to be adjusted when the correct statistical analysis is performed.

Line 212 - Edit this sentence.

Table 2 - Significance means that the environmental conditions among the tanks are different? Or that spawning/fecundity is different in those tanks?
There seems to be some error or inconsistencies with pH, because the means in system 3 and 4 are the same as 1 and 2 – the difference might have to do with the combined effect of environmental parameters. Although it is unlikely that this study can shed light on environmental parameters because the study was not specifically designed to test this. The first statistical question to address is whether the tanks even differed across time, tank number, water system, and sex ratio. If there is a difference there, then you can go on to use environmental parameters to explain the differences.

Figures: they will likely all need to change when the new analysis is conducted. But here are some comments.
Fig 1 - The line style is inappropriate for the data and collection methods. Shading in the line like this suggests that area under the curve is important, but that is not accurate. A bar graph for date and eggs collected is more appropriate.
Fig 2 - Female size is known to have an effect on fecundity. This should be included as a covariate in this analysis.
Fig 3 - M:f ratio should be per tank not per system (unless there is a valid reason for reporting all results as per system, which needs to be clearly outlined in the manuscript). Only show if there is a significant difference in the statistical analysis.
Fig 6 - Mean pH should be for the tank, not across tanks. The black line hints that time is an important variable in fecundity. See comment to Fig 1 about the figure style.

Raw Data presentation:
In the raw data, please give an overview of the conditions of each tank, and make the information more user friends. For example, how many total tanks are there? (10, according to “final Wt(g)& FL(cm)” -so in “volume of eggs and Fecundity” there should be 10 tanks, not just 5 represented, and there are 5 systems (5 pairs of tanks). In sheet “final Wt(g)& FL(cm)”, how many females are in tank 5a compared to 4b? These would be better suited as columns within data sheets – for example, add a column for the number of ratio of males to females, a column for total number of fish, a column for system number, and anything else that would be useful. You want to make the data easily understandable so that folks can use it if they need to in the future.
Please also make sure to include all the raw data, and a clear explanation for each sheet – for example what are the individual egg diameter, what are the 2 systems – see comment above but was this conducted at one timepoint in one tank system? Is this a good representation of all the fish egg sizes?

Reviewer 1 ·

Basic reporting

MS is correctly prepared and formatted. References are properly selected. The hypothesis should be corrected for, e.g.: it is possible to spontaneously spawning of French grunt in captivity, and then verified.

Experimental design

This is the biggest weakness of MS. Four fish were kept in each pool - but in some pools there were only females (no males). This factor may interfere with the reported results because it is unknown how the presence of males and their pheromones affects egg production and their ovulation and subsequent release into water.
It is not entirely clear whether the eggs were collected collectively from pools A and B (and probably it was unfortunately so) or separately. In the latter case, it would be easier to interpret the results. One could then consider individual subgroups, e.g. 2: 2 (males: females) - this is one group with repetitions; 1: 3 - is the second group; 0: 4 - this is the third group.
There is also no data on the weight of females in MS (although the text shows that the authors have these data) - then relative fecundity could be calculated, not just total. It would be easier then to present and discuss the results.
The work does not show whether the fertilization rate was tested. If so, adding this data would greatly increase the value of MS.

Validity of the findings

See "Experimental design" notes. The value of the work would also increase the information about how many days eggs were obtained from the same pools, which will allow to conclude more accurately about the frequency of reproduction. The fact that spawning at night should also be discussed more strongly. Coral reef fishes often spawn at night or their offspring (like clownfish) hatch at night. It is a very interesting biological phenomenon, especially when compared to freshwater fish.

Additional comments

This is a very difficult decision. I have read MS many times and I think that authors should be given a chance to improve their work, because of its importance for the protection of coral reef biodiversity. The practical aspect of MS is also important.

Reviewer 2 ·

Basic reporting

The manuscript submitted for review contains very important information regarding reproduction under controlled conditions of one of the coral reef fish species. This is the aquaculture topic that will be intensively developed in the near future. The manuscript is properly constructed. But my objections concern the purpose of research, hypothesis and justification for establishing research groups.

Experimental design

1. There is no research hypothesis. In the case of these studies, it is very easy to make such a hypothesis, e.g. it is possible to breed French grunt under controlled conditions.
2. The division of fish into research groups is unclear. Indeed, the authors inform that the fish was selected in terms of their similar size and the least aggression, which in coral reef fish can lead to a drastic reduction in stocking. However, different reproductive groups were created, where the ratio of males to females was 1: 1, 1: 3 and 0: 4 (so-called negative control) and they should be analyzed this way.

Validity of the findings

The results should be presented a little differently.
1. The effects of reproduction (harvested eggs) should be presented in relation to reproduction groups 1: 1, 1: 3 and 0: 4.
2. It is necessary to get eggs from individual tanks every how many days to determine the reproduction frequency. It would be interesting to study (calculate) the relationship between temperature and reproduction frequency - here each result is interesting.
3. It would be good if the results obtained were recalculated depending on the weight of the females: in this way relative fecundity would be obtained.
4. It would be very interesting to obtain data on the total fecundity (number of eggs) from individual groups throughout the duration of the study.
5. It is also important to indicate whether the fertilization percentage was calculated or not!

Additional comments

If the authors have output from individual pools, after re-calculating, e.g. relative and total fecundity, this will be a high-quality manuscript.

---

## Round 0.2 · Minor Revisions

Thank you for addressing the comments made by myself and the reviewers, this updated version is clear and nearly ready for publication.

There are a few places in the manuscript where there are extra periods, commas or spaces which will need to be edited.

While the statistical methods and results sections are vastly improved, I still think some edits for both clarity and correctness are required. And perhaps some additional/replacement analyses need to be conducted.
To be consistent, use similar terminology for your statistical tests. if you are performing a linear mixed-effects model and an ANOVA, call the ANOVA a linear model. If you are only performing ANOVAs/linear models, using the term ANOVA is fine. Here you have used the terms linear mixed effects model, linear regression analysis and ANOVA - but linear regression analysis and ANOVA are the same test - so call them the same thing.

Ideally, all analyses should be performed in the same statistical program - it keeps everything a little more streamlined and easier to follow. If that is not possible that is okay, but Excel isn't a statistical program and really should not be used as such. In lines 188-190, are you saying you performed ANOVAs and post hoc analyses in Excel? 'lm' is a base package of R, so it would be better to perform them there. Also please cite R and the lme4 package.

Comments on your linear mixed effects model investigating the effects of environmental parameters.

If you are conducting a linear mixed effects model there needs to be a random effect - typically with the lme4 package, the model won't run unless you put in a random effect. Please make sure to note what your random effects are (likely tank or system). If you didn't include a random effect then you didn't perform a linear mixed-effects model, but a linear model. And then also please explain why you didn't include tank or system as an effect because it likely is a covariate. You could instead include tank as a fixed effect and run a linear model. You then should perform post hoc tests for the variables that affect egg release.

You did not state in the analysis section that you performed post hoc analyses for the linear mixed effects model, but it seems like perhaps you had performed post hoc tests (according to table 2)

If there was no difference in most of the environmental variables then you don't need to perform post hoc analyses (table 2) - just provide the results for pH. However, Table 2 is confusing because in the statistical methods you say you performed mixed-effects models not ANOVAs for the environmental parameters. And system was not included in the analysis according to how it is written in the statistical analysis section. Please clarify in the manuscript.

Comments on the linear regression analysis investigating other effects on egg numbers:

Did you perform just one ANOVA or multiple ANOVAs for this analysis? In your results however, it seems like you performed different ANOVAs for each independent variable, so that you could get R^2 values. This is not good statistical technique (which I pointed out in my comments to the previous draft), and you should have only performed one model that included all of these variables as fixed effects.

Please clarify exactly what the variables where. The dependent variable was number of eggs, but what were the independent variables/fixed effects? If any environmental condition found to affect the number of eggs was used then only pH should have been included - please make it clear even if you are “giving away” some of your results. You can say something like “pH was included because it was an environmental condition shown to affect egg release according to the model above". Further, as stated above, system or tank should be included in your model either as a fixed effect or the model should be a linear mixed effect with system as a random effect.

ANOVA for significance between systems.

I'm not really sure what this means - what are the dependent and independent variables? The tank/system is an important variable and really should be included in your 2 models above. If there was a reason why it wasn't included, then you really need to explain why not.

The manuscript itself is clear and important. But the statistical section needs to be more robust and model parameters need some clarity. However, if there is a strong reason why you have chosen to analyze the data in a certain way (once you have clarified exactly what you did), simply state the reason in the manuscript. My above comments are because it is unclear what analyses were performed as the manuscript stands, and based on my assumptions given how the results are laid out.

Thank you for your effort, this manuscript is nearly ready for publication.

Reviewer 1 ·

Basic reporting

MS has been improved in line with my comments. I have no negative comments about the current version.

Experimental design

The MS after corrections, including the description of the experiment, are presented correctly, legibly and clearly.

Validity of the findings

No comment

Additional comments

In my opinion, the current version of MS is absolutely acceptable for printing.

Reviewer 2 ·

Basic reporting

The paper has been very well improved. My comments have been taken into account. Paper is easy to read now, is understandable and clear.

Experimental design

No comments

Validity of the findings

No comments

Additional comments

The paper, after corrections, is of high quality. It refers to very important reproduction issues of coral reef fishes.

---

## Round 0.3 · accepted · Accept

I'm happy with these edits to the statistical analyses. Thank you for making it much clearer.